# Fabaceae: South African Medicinal Plant Species Used in the Treatment and Management of Sexually Transmitted and Related Opportunistic Infections Associated with HIV-AIDS

**Nkoana Ishmael Mongalo [1,\*] and Maropeng Vellry Raletsena [1,2]**

1   CAES Laboratories, College of Agriculture and Environmental Science, University of South Africa, Private Bag x06, Florida Park 0710, South Africa; raletmv@unisa.ac.za
2   Department of Agriculture and Animal Health, College of Agriculture and Environmental Science, University of South Africa, Private Bag x06, Florida Park 0710, South Africa
\*   Correspondence: mongani@unisa.ac.za; Tel.: +27-116709445

**Abstract:** The use of medicinal plants, particularly in the treatment of sexually transmitted and related infections, is ancient. These plants may well be used as alternative and complementary medicine to a variety of antibiotics that may possess limitations mainly due to an emerging enormous antimicrobial resistance. Several computerized database literature sources such as ScienceDirect, Scopus, Scielo, PubMed, and Google Scholar were used to retrieve information on Fabaceae species used in the treatment and management of sexually transmitted and related infections in South Africa. The other information was sourced from various academic dissertations, theses, and botanical books. A total of 42 medicinal plant species belonging to the Fabaceae family, used in the treatment of sexually transmitted and related opportunistic infections associated with HIV-AIDS, have been documented. Trees were the most reported life form, yielding 47.62%, while *Senna* and *Vachellia* were the frequently cited genera yielding six and three species, respectively. *Peltophorum africanum* Sond. was the most preferred medicinal plant, yielding a frequency of citation of 14, while *Vachellia karoo* (Hayne) Banfi and Glasso as well as *Elephantorrhiza burkei* Benth. yielded 12 citations each. The most frequently used plant parts were roots, yielding 57.14%, while most of the plant species were administered orally after boiling (51.16%) until the infection subsided. Amazingly, many of the medicinal plant species are recommended for use to treat impotence (29.87%), while most common STI infections such as chlamydia (7.79%), gonorrhea (6.49%), syphilis (5.19%), genital warts (2.60%), and many other unidentified STIs that may include "Makgoma" and "Divhu" were less cited. Although there are widespread data on the in vitro evidence of the use of the Fabaceae species in the treatment of sexually transmitted and related infections, there is a need to explore the in vivo studies to further ascertain the use of species as a possible complementary and alternative medicine to the currently used antibiotics in both developing and underdeveloped countries. Furthermore, the toxicological profiles of many of these studies need to be further explored. The safety and efficacy of over-the-counter pharmaceutical products developed using these species also need to be explored.

**Dataset:** https://www.mdpi.com/article/10.3390/plants11233241/s1

**Dataset License:** Attribution 4.0 International (CC BY 4.0)

**Keywords:** South African traditional medicine (SATM); ethnobotany; gonorrhea; syphilis; chlamydia; impotence; fungal infections; *Mycoplasma* species

## 1. Summary

South African traditional medicine (SATM) involves the use of medicinal plants, animal fats and bones, birds, and stones from various sources [1]. Although these plants may not be well studied, they could well serve as an alternative and complementary medicine

to modern antibiotics [2]. The development of antimicrobial resistance (AR) to many of the commonly used antibiotics in developed, underdeveloped, and developing countries worldwide further supports and warrants the incorporation of the use of medicinal plants in treating a variety of devastating human illnesses, particularly sexually transmitted and related infections [3]. Sexually transmitted infections (STIs) primarily spread through various types of sexual contact, including unprotected sexual intercourse. These increase the chances of HIV infection through semen and vaginal fluids [4]. As there is no cure yet, HIV progresses and result in a plethora of opportunistic infections, the further loss of immunity, and possible hospitalization, which would lead to excessive financial costs [5]. STIs may well be caused by a variety of microorganisms of different origins, which may be resistant to many common antibiotics (Table 1).

**Table 1.** Types and names of sexually transmitted infections common worldwide.

| Type of Organism | Organism(s) Involved | Known STI(s) |
| :---: | :---: | :---: |
| **Bacteria** | *Neisseria gonorrhoeae* | Gonorrhoeae |
| | *Chlamidia trachomatis* | Chlamydial infection |
| | *Klebsiella granulomatis/Granuloma inguinale* | Donovanosis |
| | *Mycoplasma hominis, Mobiluncus* spp | Bacterial vaginitis |
| | *Mycoplasma genitalium* | Urogenital infections |
| **Parasites** | *Trichomonas vaginalis* | Vaginal trichomoniasis |
| **Yeasts** | *Candida albicans* | Vulvovaginitis |
| **Viruses** | Cytomegalovirus | Inflammation of bowel |
| | HSV type 2 | Genital herpes |
| | HSV-8 | Kaposi's sarcoma |
| | Hepatitis B | Hepatitis |
| | Human immunodeficiency virus | HIV/AIDS |

It is also important to note that the rate at which antibiotics are discovered is extremely slow, hence the use of medicinal plants is critical. Fabaceae is the second largest family of medicinal plants, containing 751 genera and approximately over 19,500 plant species, most of which have been used as traditional medicines and are mainly trees, shrubs, and perennial or annual herbaceous plants, which are easily recognized by their distinct fruit (legume) [6,7]. Medicinally, the species are used in the treatment of a variety of devastating human and animal infections, ranging from pain and inflammation, tuberculosis, infertility, STIs, mastitis, and many other life-threatening illnesses [8,9]. In the current work, South African medicinal plants from the Fabaceae family, used in the treatment of sexually transmitted and related infections, are documented from various search engines such as Google Scholar, PubMed, Web of Science, and Science direct, as well as other sources such as books and various theses and dissertation documents from various universities within the country. Although plenty of medicinal plants in South Africa are reported to treat STIs and related infections, it is important to note that there is little evidence, particularly on the in vitro antimicrobial activity of these plants against a plethora of microorganisms implicated as causative agents of STIs [10]. Furthermore, in vivo and toxicological aspects of these plants need to be explored. This is imperative as many of these reported medicinal plants are major ingredients of medicines sold over the counter in many unregistered pharmacies within South Africa. This is a major setback to traditional medicine in Africa at large. It is also important to further explore the phytocompounds, individually or synergistically, against major causative agents of STIs.

## 2. Data Description

Medicinal plants from the Fabaceae family, used in the treatment of sexually transmitted and related infections, are summarized in Table 2, with columns showing the plant species (scientific names) matched using the Plantzlist, National Biodiversity Institute (NBI) to comply with both national and international standards [10]. Furthermore, to avoid

using synonyms of the medicinal plants documented and to verify the authors' names, https://powo.science.kew.org/ (accessed on 24 August 2023) was also used. Various plant parts used and indigenous names (various vernacular names notated in letters: Z—Zulu; X—Xhosa; Tsh—Tshivenda; S—Sotho; and E—English), mode of preparation, indications of specific use and the number of citations from the literature were also reported.

**Table 2.** Fabaceae plant species used in the treatment of sexually transmitted and related infections in South Africa.

| Plant Species | | Plant Parts Used, Indigenous Names | Mode of Preparation | Therapeutic Indications and Reference(s) | Total Number of Citations |
|---|---|---|---|---|---|
| *Albizia adianthifolia* (Schumach.) W.Wight var. *adianthifolia* | T | Leaves; Igowane (Z); Flat-crown (E); Isicangca (X); Muomba-ngoma (Tsh) | Boiled and taken as an enema | Syphilis and impotence [11,12] | 2 |
| *Albizia anthelmintica* (A.Rich.) Brongn. | SH | Roots; Muime (Tsh); Cherry-blossom tree (E); Mmola (S); Umnala (Z) | Not mentioned | Various unidentified STIs [13] | 1 |
| *Albizia gummifera* (J.F.Gmel.) C.A.Sm. | T | Stem bark; Umgandakawu (Z) | Boiled and taken orally | Various unidentified STIs [14,15] | 2 |
| *Albizia versicolor* Welw. ex Oliv. | T | Stem bark; Muvhambangoma (Tsh); Mmola (S); Mohlalabata (S); Umphisu (Z) | Not mentioned | Various unidentified STIs, impotence in men [13,16]. | 2 |
| *Bauhinia galpinii* N.E.Br. | SH | Roots; Mohohoma (S); Mutswiriri (Tsh) | Not mentioned | Various unidentified STIs, impotence [12,17,18]. | 3 |
| *Bolusanthus speciosus* (Bolus) Harms | SH | Roots and stem; Mukambana (Tsh) | Boiled and taken orally | Various unidentified STIs and impotence [12,13,18–20] | 5 |
| *Burkea africana* Hook. | T | Roots and seeds; Monatlo (S), Mufhulu (Tsh) | Both roots and seeds are ground and boiled and drunk | Opportunistic infections associated with HIV-AIDS [12,21–24] | 5 |
| *Caesalpinia decapetala* (Roth) Alston. | SH | Roots; Mokgabane (S) | Boiled and taken orally | Impotence and gonorrhea [12,24–28] | 6 |
| *Cassia abbreviata* Oliv. | SH | Roots and stem bark; Monepenepe, Molomanama (S) | Boiled and taken orally | Various unidentified STIs, chlamydia, impotence, and as an immune booster for HIV-AIDS patients [12,17,21,29–32] | 7 |
| *Colophospermum mopane* (J.Kirk ex Benth.) J.Léonard | T | Roots; Mopane (S) | Boiled and taken orally | Impotence [12] | 1 |
| *Dalbergia melanoxylon* Guill. and Perr. | | Stem bark; African Blackwood (E); Ebony (E); Grenadille Wood (E); Muuluri (Tsh) | Not mentioned | Various unidentified STIs [16] | 1 |
| *Dichrostachys cinerea* (L.) Wight and Arn. | SH | Roots; Moretshe (S) | Not mentioned | Impotence and syphilis [12,21,33] | 3 |
| *Elephantorrhiza burkei* Benth. | H | Roots; Mohauwane wa thaba, Mositsane (S), Tshisevhufa (Tsh) | Applied to infected areas of the skin, decoction drunk | Various unidentified STIs, chlamydia, aphrodisiac, and opportunistic infections associated with HIV-AIDS [12,13,17–19,21,22,24,32,34–36] | 12 |
| *Elephantorrhiza elephantina* (Burch.) Skeels | H | Roots; Mohauwane, Mosehlana, Mositsane (S) | Boiled and taken orally | Syphilis, chlamydia, impotence in men [17,21,23,32,33,35–40] | 11 |
| *Erythrina caffra* Blanco | SH | Roots; Umsisnsi | Boiled and taken orally | Various unidentified STIs and genital warts [11,15] | 2 |
| *Eriosema cordatum* E.Mey. | H | Roots; Leshetla (S); Ubangalala (Z); Umhlabankunzi (Z); Uqonsi (Z); Uqontsi (Z) | Not mentioned | Impotence in men [39] | 1 |
| *Erythrina lysistemon* Hutch. | T | Stem bark; Muvhale (Tsh) | Boiled and taken orally | Impotence in men [18,22] | 2 |
| *Faidherbia albida* (Delile) A.Chev. | T | Stem bark; Muhoto (S) | Not mentioned | Various unidentified STIs and impotence for men [12,13] | 2 |

**Table 2.** *Cont.*

| Plant Species | | Plant Parts Used, Indigenous Names | Mode of Preparation | Therapeutic Indications and Reference(s) | Total Number of Citations |
|---|---|---|---|---|---|
| *Lessertia depressa* Harv. | SH | Roots, Mmusapelo (S) | Decoction taken orally | To treat vaginosis [41] | |
| *Melolobium microphyllum* (L.f.) Eckl. and Zeyh. | SH | Whole plant, Mofahlatoeba (S) | Decoction taken orally | To treat unidentified STIs [41] | |
| *Mundulea sericea* (Willd.) A.Chev. | SH | Roots; Maibana, Mohato, Mositatlou (S); Mukandandou (Tsh) | Not mentioned | Impotence in men [12,13] | 2 |
| *Ormocarpum trichocarpum* (Taub.) Engl. | T | Roots; Caterpillar Bush (E); Caterpillar Pod (E); Hairy Caterpillar Pod (E); Umsindadlovana (Z); Mosepe (S); Mugogodwane (Tsh), Muthari (Tsh); | Not mentioned | Impotence in men [42] | 1 |
| *Peltophorum africanum* Sond. | T | Roots and stem bark; Mosehla (S); Musese (Tsh) | Boiled and taken orally | Various unidentified STIs, impotence, and skin infections associated with HIV-AIDS [11,13,16,17,19, 22,24,25,27,35–38,41,43] | 14 |
| *Piliostigma thonningii* (Schum.) Milne-Redh. | T | Roots; Mukolokote (Tsh); Camel's Foot (E); Rhodesian Bauhinia (E); Mokgôrôpô (S); | Not mentioned | Various unidentified STIs and impotence [13,42] | 2 |
| *Philenoptera bussei* (Harms) Schrire | T | Roots, Mphata (S) | Boiled and taken orally | Chlamydia [31] | 1 |
| *Philenoptera violacea* (Klotzsch) Schrire | T | Roots; Apple-leaf (E) | Not mentioned | Impotence in men [12] | 1 |
| *Pterocarpus angolensis* DC. | T | Stem bark; Mutondo (Tsh) | Not mentioned | Various unidentified STIs and "*divhu*" [12,13,19,21,44] | 4 |
| *Pterocarpus rotundifolius* (Sond.) Druce | T | Stem bark; Muataha (Tsh) | Boiled and taken orally | Oral candidiasis associated with HIV-AIDS [22] | 1 |
| *Schotia brachypetala* Sond. | T | Roots and stem bark; Mununzu (Tsh); African Greenheart (E); African Walnut (E); Uvovovo (Z) | Boiled and taken orally | Opportunistic infections associated with HIV-AIDS [22,45] | 1 |
| *Senegalia ataxacantha* (DC.) Kyal. and Boatwr. | T | Roots; Muluwa (Tsh); Flame Acacia (E); Flame Thorn (E); Mogokare (S), Ubophe (Z); Ugagane (Z), Umnga (X) | Not mentioned | Impotence in men [13] | 1 |
| *Senegalia caffra* (Thunb.) P.J.H.Hurter and Mabb. | T | Leaves; Morobadiepe (S), Murhovhambado (Tsh); Muvunda-mbado (Tsh); Morobadiepe (S); Umnyamanzi (X); Umthole (X); White thorn (E) | Leaves are dried, burned, and mixed with animal fat, applied to mouth blisters and ulcers | Opportunistic infections associated with HIV include mouth ulcers [22] | 1 |
| *Senna auriculata* (L.) Roxb. | T | Leaves; Muduwishango (Tsh) | Fresh leaves are crushed, immersed in water, and drunk orally | Opportunistic infections associated with HIV-AIDS [22] | 1 |
| *Senna didymobotrya* (Fresen.) H.S.Irwin and Barneby | SH | Roots; African senna (E); popcorn senna (E); candelabra tree (E); peanut butter cassia (E) | Crushed and boiled, taken orally | Chlamydia, gonorrhea, syphilis, aphrodisiac, and genital warts [12,24–26,45] | 5 |
| *Senna italica* Mill. | H | Roots; Morotelatshotshi, Setlommana, Mankgane (S) | Boiled and taken orally | Various unidentified STIs, impotence, gonorrhea, "*makgoma*" and opportunistic infections associated with HIV-AIDS vaginal candidiasis [22,23,25,27,30,32,33,35] | 9 |
| *Senna occidentalis* (L.) Link | SH | Roots; Modulabadimo (S) | Not mentioned | Impotence in men [12] | 1 |
| *Senna petersiana* (Bolle) Lock | SH | Seeds, Munembenembe (Tsh); Bohlôko (S); Dwarf Cassia (E); Umnembenembe (Z) | Dried seeds are ground into powder and boiled in water and drunk | Impotence, gonorrhea, and opportunistic infections associated with HIV-AIDS [12,13,21,22,43–46] | 8 |

| Plant Species | | Plant Parts Used, Indigenous Names | Mode of Preparation | Therapeutic Indications and Reference(s) | Total Number of Citations |
|---|---|---|---|---|---|
| *Senna sophera* (L.) Roxb. | SH | Leaves; Mutsheketsheke (Tsh) | Leaves are boiled and taken orally three times a day | Gonorhea [21] | 1 |
| *Sutherlandia frutescens* (L.) W.T.Aiton | SH | Leaves; Mokgoroma (S) | Boiled and taken orally | Opportunistic infections associated with HIV-AIDS [43] | 1 |
| *Tephrosia zoutpansbergensis* Bremek. | SH | Roots; Motswaing (S) | Not mentioned | Impotence in men [12] | 1 |
| *Vachellia karroo* (Hayne) Banfi and Glasso | T | Roots and stem bark, Mooka (S), Muunga (Tsh); Sweet thorn (E); Soetdoring (A); | Boiled, used as a mouthwash, or applied directly to the genitals to treat vaginal ulcers. | Various unidentified STIs, impotence, opportunistic infections associated with HIV vaginal candidiasis and ulcers [12,13,19–22,37,47–50] | 12 |
| *Vachellia permixta* (Burtt Davy) Kyal. and Boatwr. | SH | Roots; Moselaphala (S); Hairy Acacia (E); Mimosa (E) | The decoction is taken orally | Chlamydia [31] | 1 |
| *Vachellia robusta* (Burch.) Kyalangalilwa and Boatwright subsp. *robusta* | T | Stem bark and roots; Brack Thorn (E); Moga (S); River Thorn (E); Umngamanzi (Z); Umngampunzi (X) | Not mentioned | Impotence in men and opportunistic infections associated with HIV-AIDS [12,51] | 2 |

The Fabaceae family from a plethora of countries have been reported to be dominant in the treatment of many human and animal infections [52]. According to Asfaw and Abebe [53], the family comprises many plant species with diverse characteristics, which makes it one of the most important families to humans and is used as medicine, crops, green manures, and forages. From the 42 medicinal plant species documented, most of the plant species are trees (47.62%), followed by shrubs (42.85%) and herbs (9.52%) as shown in Figure 1. Elsewhere, the dominance of the use of trees and shrubs has been widely reported [54,55]. *Senna* was the most represented genus, recording six plant species, while *Albizia* recorded three species, and the other five genera such as *Senegalia*, *Philenoptera*, *Erythrina*, *Pterocarpus*, and *Elephantorrhiza*, recorded two plant species each (Figure 2). According to Oladeji et al. [56], members of the genus *Senna* are important in African traditional medicine and are used to treat a variety of devastating human and animal ailments, and yields some important pharmacological activities that includes antimicrobial, antipyretic, antimalarial, antidiabetic, anti-inflammatory, and antiproliferative activities. Over and above the biological activities reported, the species may well regulate oxidative stress in many cardiovascular diseases [57]. The major phytocompounds of the genus mainly include volatile oils, glycosides, tannins, terpenoids, anthraquinones, saponins, tannins, flavonoids, and a plethora of alkaloids, which may well account for their diverse uses in traditional medicine [58].

Although *Senna* species are dominant, it is important to note that in traditional medicine, plants are combined and taken orally until the infection, particularly sexually transmitted infection, is healed [48]. The combined version of plants is believed to prevent microbial resistance and further purge the disease. However, the role of purgatives has not been well studied scientifically. With plant species reported in the current study, *P. africanum*, *E. burkei*, and *Cassia abbreviata* may be boiled to treat many unidentified STIs and a plethora of opportunistic infections associated with HIV-AIDS [59]. *Peltophorum africanum* recorded the highest number of citations (14), followed by *Vachelia karroo* and *Elephantorhiza burkei* with 12 and *Elephantorhiza elephantina* with 11 citations (Figure 3). According to Maroyi [60], the leaves, roots, and stem bark from *P. africanum* are generally used to treat syphilis in Zimbabwe, while a stem bark from *P. africanum* is used to treat gonorrhea and some opportunistic infections associated with HIV-AIDS [51]. These results corroborate our results on the dominance of the use of plant species in treating STIs across

a variety of cultures and countries. Although the plant species is dominant, the scientific validation of the species against a plethora of causative agents is lagging and still needs to be explored from in vitro and in vivo to clinical trials.

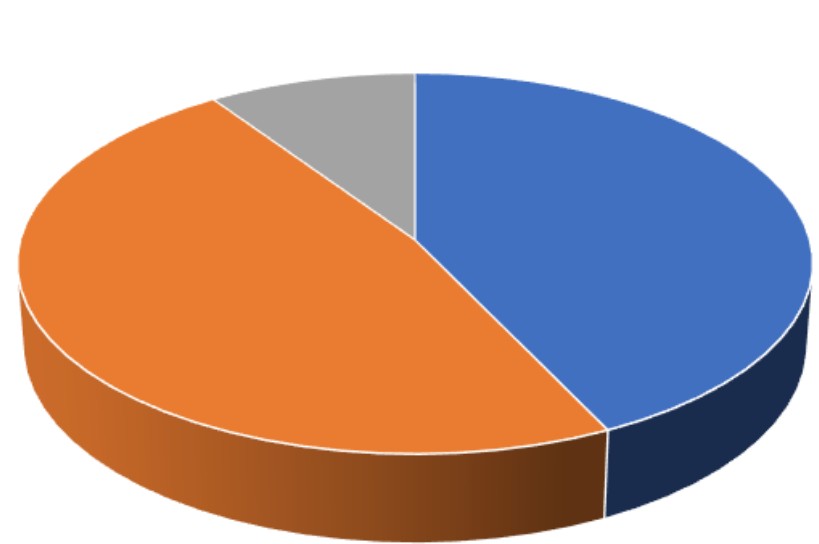

**Figure 1.** Different forms of the family used to treat sexually transmitted and related infections.

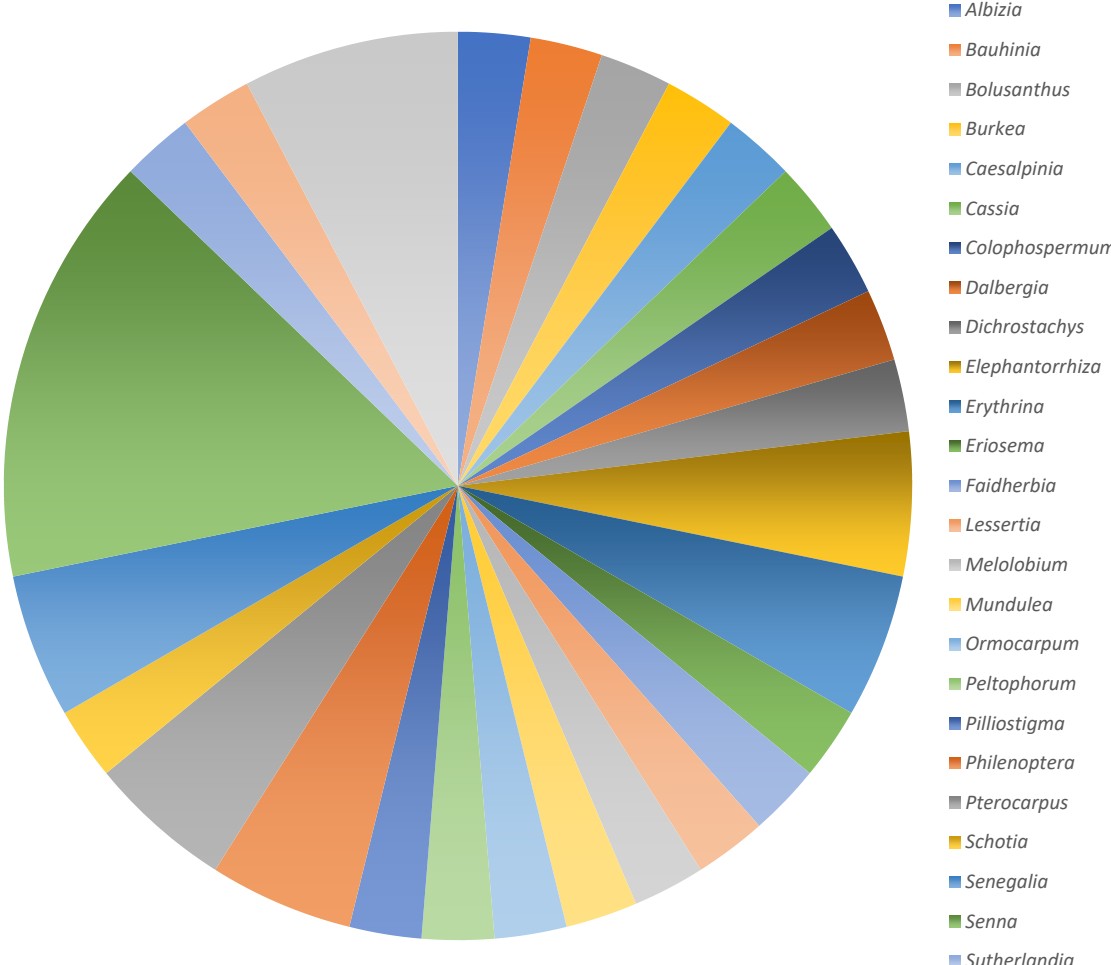

**Figure 2.** Different numbers of species per genus, used to treat STIs and related infections.

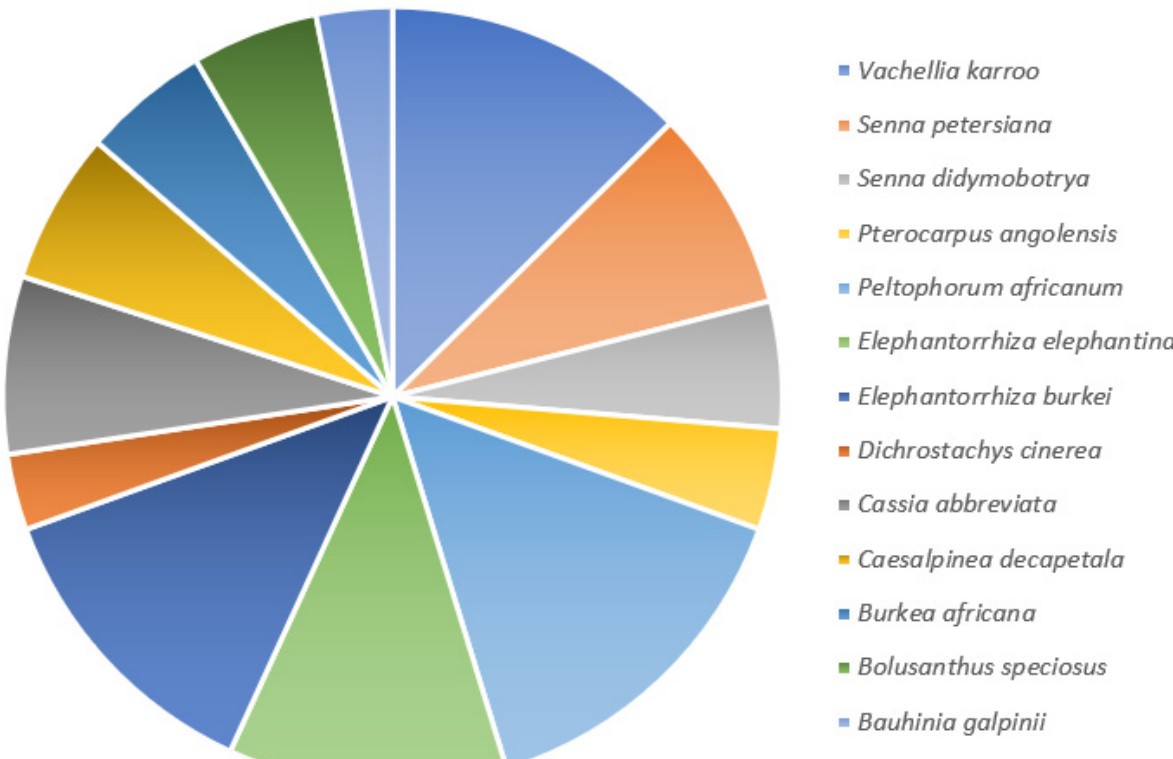

**Figure 3.** The Fabaceae plant species used in the treatment of STIs and related infections, with a higher number of citations (>2).

Most of the plant parts used from the Fabaceae species were roots at 57.14% followed by stem bark at 27.53%, and leaves (10.20%), seeds (4.04) and whole plant at 2.04%, respectively, as shown in Figure 4 below. Although the dominant plant parts used may possess a plethora of significant secondary metabolites that may well explain the choice of the plant part, it is important to note that the use of roots and stem bark may be detrimental to the health of the plant species, leading to either the extinction or depletion of both species' diversity and richness. In Southern Africa, many of the documented plant species are drastically reduced in their natural environment [27,29,31,43]. Some species, particularly trees, are used for other purposes that may include firewood, building huts, fences, kraals, and hunting. For example, *Cassia abbreviata* stem bark is used in the Tsonga culture to cook the meat of an animal that was killed while hunting [61]. This is believed to provide the hunters with a stronger chance of killing the other animals, which are used as meat that alleviates hunger in many communities.

In the mode of administration of the plant species, most of the plant species are boiled and taken orally (51.16%), while 37.21% of the species do not mention mode of administration (Figure 5). This may well explain how sacred the traditional healers and traditional folks treat their knowledge. However, in most African cultures, sexuality is a strict and difficult topic that is less likely to be discussed publicly [62]. To a lesser extent, the documented medicinal plants are administered as mouth wash/blisters (4.65%), enema, applied directly to the affected skin and when the plant species are infused in water. It is important to note that both immersion and boiling of plants is made using water as a solvent. In our recent study, higher amounts of heavy metals were detected in plant-based medicines prepared in water with an intention of treating STIs, particularly impotence [63].

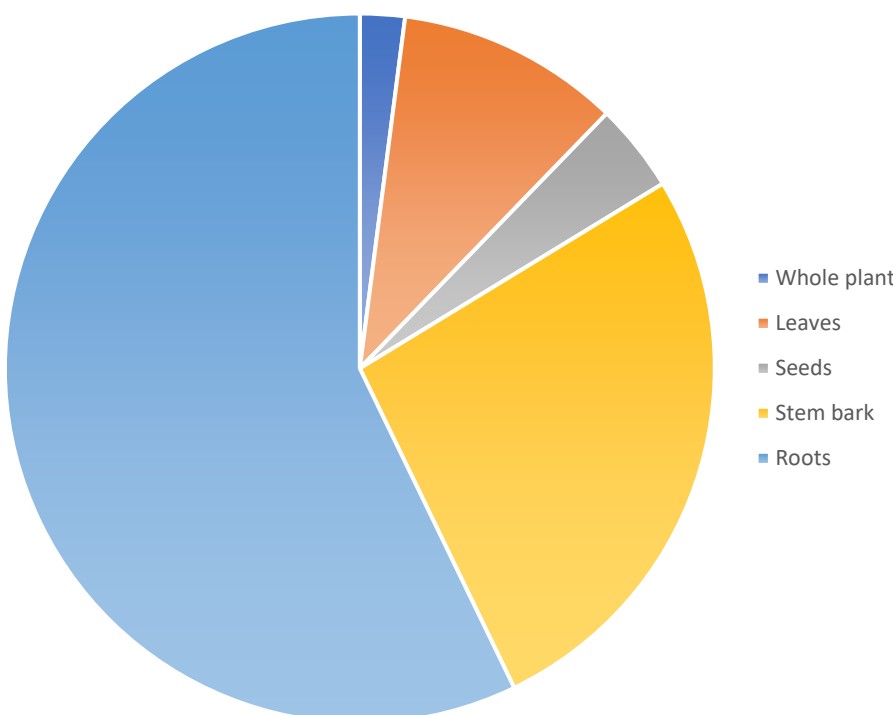

**Figure 4.** Plant parts used from Fabaceae plant species used in the treatment of STIs and related infections.

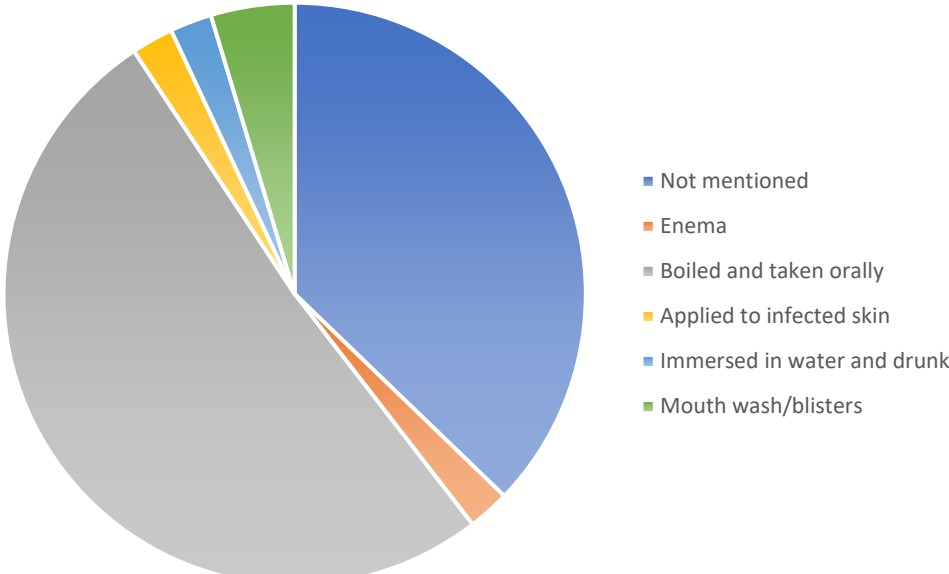

**Figure 5.** Mode of administration of the Fabaceae plant species used in the treatment of STIs and related infections.

Amazingly, the highest % of the Fabaceae plant species are used in the treatment of impotence (29.87%), followed by unidentified STIs (25.97%) and opportunistic infections associated with HIV-AIDS (18.18%), while the lower percentages were recorded for infections such as genital warts and "divhu" and "makgoma" at 2.60%, respectively (Figure 6). Divhu is a TshiVenda word that means "go wela" in Sepedi, meaning the male individual would have slept with a woman who had an unidentified STI or was on her period [17], which then made him sick. In most instances, it specifically refers to a situation where a woman was on her period or had miscarried an unborn baby and had intercourse.

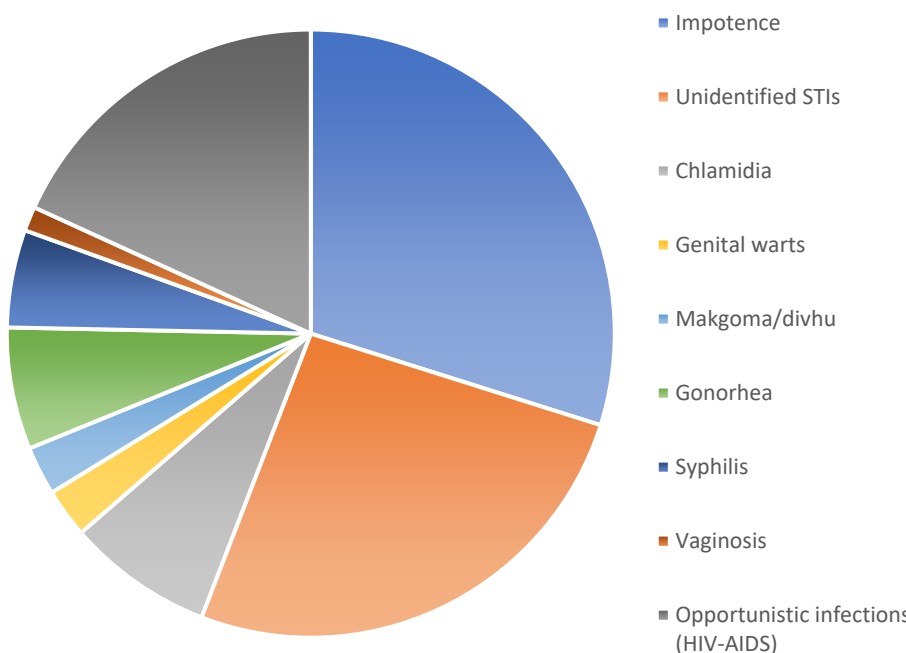

**Figure 6.** Diseases treated using the Fabaceae medicinal plants in South Africa.

Many authors have reported syphilis and gonorrhea to be highly prevalent in Africa [64–67]. In the current work, the data reported support the indigenous use of the various Fabaceae plant species used in the treatment of sexually transmitted and associated opportunistic infections associated with HIV-AIDS from various cultures and ethnic groups. Some of the species are also used for a similar purpose in other sub-Saharan countries showing a wider cross-cultural use [49,60].

## 3. Methods

### 3.1. Strategy for Literature Search

The information contained in the current work was collected as part of a literature search in various computer databases such as ScienceDirect, Scopus, Scielo, Scifinder, PubMed, Web of Science, and Google Scholar. Additional information was also obtained from various academic dissertations, general plant sciences, ethnomedicine, and other relevant ethnobotanical books. This was performed according to the guidelines of the Preferred Reporting Items for Systematic Reviews and Meta-analyses (PRISMA) statement (from 1990 to December 2023) [68]. Key words such as South Africa, medicinal plants, sexually transmitted infections, opportunistic infections associated with HIV-AIDS, ethnomedical applications, survey, and ethnopharmacological aspects were used interchangeably. The limitation of the current study lies primarily in the exclusion of the plant species traditionally used to treat tuberculosis only.

### 3.2. Data Mining

To create the data, the inclusion criteria were the following: (1) the literature has an ethnobotanical or ethnopharmacological context and the articles should be ethnobotanical field studies/surveys reporting on one or more plants with an indication of use for the treatment of sexually transmitted infections (STIs) and related diseases; (2) the place of study must be South Africa; (3) the study must focus on plants; and (4) the coursework must be written in English. On the other hand, the exclusion criteria were as follows: (1) articles without scientific plant names; (2) review articles; and (3) papers focusing on animals and other natural resources used to treat sexually transmitted diseases and other conditions such as tuberculosis. Plant species reported to treat urinary tract infections, infertility, cancer sores with no evidence of HIV-AIDS, bladder problems, chapped lips,

as well as to increase libido and laxatives were also excluded. Plant species were further verified using SANBI (South African National Biodiversity Institute) and the website The Plant List (http://www.theplantlist.org, accessed 9 October 2022). Moreover, https://powo.science.kew.org/ (accessed on 24 August 2023) was also used to validate the authors' and species' names. Data were collected with the help of library staff at the University of South Africa (Florida campus). In the current work, plant species with only genus names were omitted from the search engine. The papers, books, and other sources used in the current work have been reviewed for inclusion and range from 1990 to 2023. Contributions that appeared as duplicates, were cited in abstract form, or were not in English were excluded (Figure 7). The task was performed by the first author and confirmed by the second author. For each of the relevant articles, the scientific names, family, parts of plants, method of preparation, and use in the treatment and treatment of sexually transmitted and related opportunistic infections were recorded.

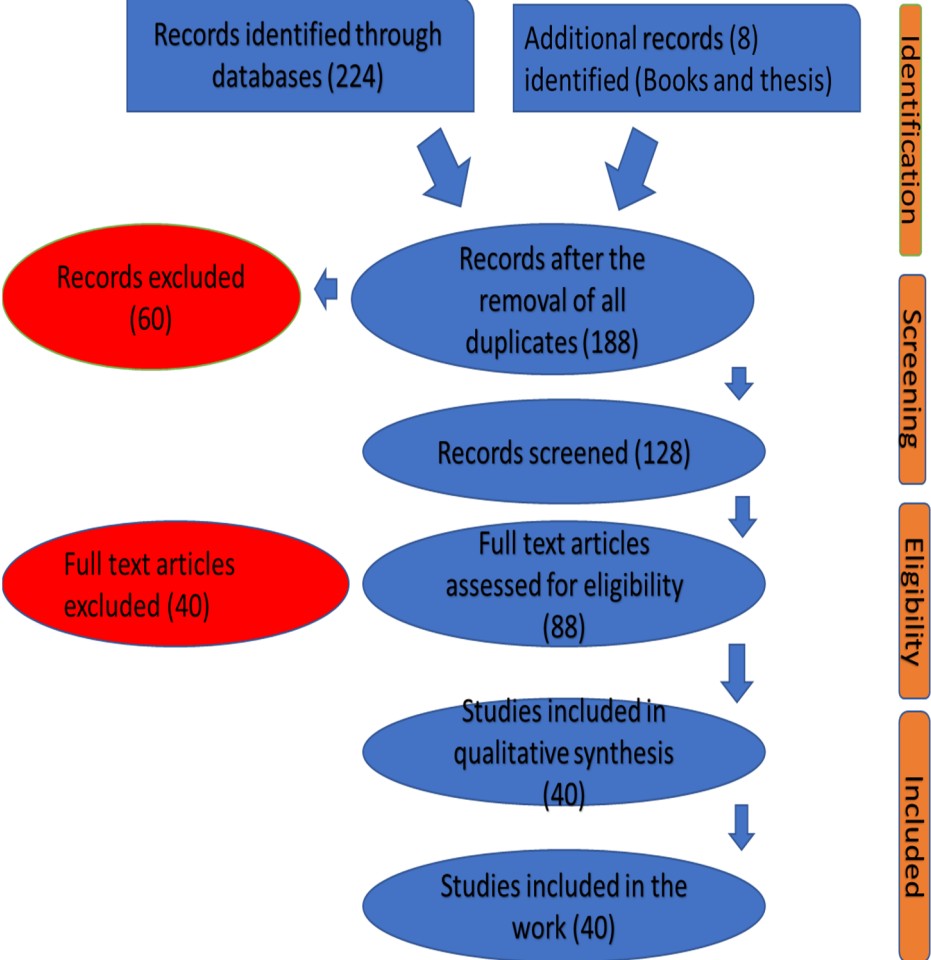

**Figure 7.** Literature sources used to collate the data of Fabaceae medicinal plants used to treat STIs and related opportunistic infections.

**Author Contributions:** Conceptualization, N.I.M. and M.V.R.; methodology, N.I.M. and M.V.R.; software, M.V.R.; validation, N.I.M.; formal analysis, M.V.R.; investigation; resources, N.I.M. and M.V.R.; data curation, N.I.M.; writing—original draft preparation, N.I.M. and M.V.R.; writing—review and editing, M.V.R.; visualization, N.I.M.; supervision, M.V.R.; project administration. All authors have read and agreed to the published version of the manuscript.

**Funding:** This research received no external funding.



**Institutional Review Board Statement:** The study was conducted in accordance with the Declaration of Helsinki and approved by the Institutional Review Board (or Ethics Committee) of University of South Africa, (Mongalo NI 90229436).

**Informed Consent Statement:** Not applicable.

**Data Availability Statement:** Data for this work are available at https://www.mdpi.com/article/10.3390/plants11233241/s1 (accessed on 25 November 2022).

**Acknowledgments:** The authors are thankful to the University of South Africa library staff for assistance with the library books and retrieval of archives and papers from various search engines identified by the researchers.

**Conflicts of Interest:** The authors declare no conflict of interest.

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
