# Peer review of "Fabaceae: South African Medicinal Plant Species Used in the Treatment and Management of Sexually Transmitted and Related Opportunistic Infections Associated with HIV-AIDS"

_data, 2023_

Round 1

Reviewer 1 Report

General Comments

The authors attempted to review medicinal plant uses of the plant family Fabaceae in the field of sexually transmitted infections in South-Africa over a relatively long time periods, which is important and commendable, particularly as this is one area of expertise of traditional healers.

However, the relation between sexually transmitted infections, opportunistic infections and HIV/AIDS needs to be clarified. What is related to what?

Categories of sexually transmitted infections should be concretized.

Gaps in the reviewed literature should be highlighted and analysed and respective recommendations for further research should be given.

Specific Comments

Some points should be clarified (please see Specific Comments)

Author Response

Good Day

Pleas find the attached. The authors have addressed all the comments as required. Thank you for your valuable comments.

Reviewer 2 Report

Dear Authors,

I found your manuscript very interesting and current. However, the form of writing is very poor and does not suit a scientific text at all. First of all, I found many errors in the scientific names. Sometimes it is a matter of writing errors, sometimes of interpretation of the taxon considered. I see that you referred to the National Biodiversity Institute and the website The Plant List: however I think it is more appropriate to verify the scientific names (avoiding synonyms) and their respective authors using https://powo.science.kew.org/

As for the contents, I think you should comment a little something about how to consider impotence among the diseases you refer to in the title.

I believe that the manuscript needs to be greatly strengthened in its Introduction section as many sentences seem self-referential or arising from a discussion that is not referred to in that context. Therefore I believe it is necessary for you to provide, where indicated by me, some appropriate bibliographical references. For your convenience I have suggested several.

Please, try to improve the figures and tables captions.

Please, check the attached file for other comments and corrections.

Best wishes.

Dear Authors,

Although I am not a native speaker, I have a lot of experience in scientific texts written in english. The form of the text as a whole is sometimes very complicated and not easy for the reader to understand: it almost seems like a literal translation from your language. I made some corrections and highlighted directly in the text some checks you need to carry out. In any case, I suggest to have a native speaker check the article.

Best regards.

Author Response

Good day.

The authors have addressed all the valuable comments from you. We hope that manuscript will add value to the Scientific body of knowledge. Thanks again. Please see the attached.
